# Genome-centric analysis of short and long read metagenomes reveals uncharacterized microbiome diversity in Southeast Asians

Jean-Sebastien Gounot[1,9], Minghao Chia[1,9], Denis Bertrand[1,9], Woei-Yuh Saw[2,3], Aarthi Ravikrishnan[1], Adrian Low [4], Yichen Ding[4], Amanda Hui Qi Ng[1], Linda Wei Lin Tan [5], Yik-Ying Teo[2,5,6,10] ✉, Henning Seedorf [4,7,10] ✉ & Niranjan Nagarajan [1,8,10] ✉

Despite extensive efforts to address it, the vastness of uncharacterized 'dark matter' microbial genetic diversity can impact short-read sequencing based metagenomic studies. Population-specific biases in genomic reference databases can further compound this problem. Leveraging advances in hybrid assembly (using short and long reads) and Hi-C technologies in a cross-sectional survey, we deeply characterized 109 gut microbiomes from three ethnicities in Singapore to comprehensively reconstruct 4497 medium and high-quality metagenome assembled genomes, 1708 of which were missing in short-read only analysis and with >28× N50 improvement. Species-level clustering identified 70 (>10% of total) novel gut species out of 685, improved reference genomes for 363 species (53% of total), and discovered 3413 strains unique to these populations. Among the top 10 most abundant gut bacteria in our study, one of the species and >80% of strains were unrepresented in existing databases. Annotation of biosynthetic gene clusters (BGCs) uncovered more than 27,000 BGCs with a large fraction (36–88%) unrepresented in current databases, and with several unique clusters predicted to produce bacteriocins that could significantly alter microbiome community structure. These results reveal significant uncharacterized gut microbial diversity in Southeast Asian populations and highlight the utility of hybrid metagenomic references for bioprospecting and disease-focused studies.

While estimates for microbial diversity on Earth vary widely, studies suggest that there are nearly a million prokaryotic species of which only around 20,000 have been cultured[1,2]. The use of culture-free metagenomic techniques has therefore been key to unravel this 'dark matter' of genetic diversity on Earth. Microbial communities in a wide-range of biospheres have been explored, including terrestrial[3], aquatic[4] and extreme environments[5], as well as plant, animal and human-associated microbiomes[6]. Improvements in metagenomic assembly workflows[7–11] and computing resources have further enabled the assembly of these large datasets to construct metagenome-assembled

genomes (MAGs) that serve to augment isolate-based reference genome databases[12,13]. Despite this, existing databases only represent approximately 48,000 species with genome sequences, and the accuracy and completeness of short-read based MAGs is frequently lower than isolate-based references[2].

Human gut metagenomes represent an area of intense scientific interest due to their association with various cancers, metabolic, immunological and neurological disease conditions[14,15]. Metagenome-wide association studies frequently rely on the completeness of reference genomes to correctly assign short reads to taxa, and link

microbial genes and function to diseases[16]. In particular, existing studies suggest that there might be key population-specific differences in metagenomic associations with various diseases[17–19]. The availability of a large number of short-read metagenomic datasets (e.g., >20,000 for human gut in public repositories) has spurred the generation of MAG reference collections based on short-read assembly[13,20–22]. While these studies have added an impressive collection of genomes to existing databases, it is unclear yet if they are representative of the genetic diversity seen in gut metagenomes around the world. In addition, recent advances in sequencing assays (e.g., Hi-C[23], read cloud[24]), hybrid[25] and long-read metagenomic analysis[26] have sought to address the shortcomings of short-read metagenomics, and opened the possibility that long-read based MAGs can provide near-complete genomes rivaling isolate genomes in quality. As access to genome sequencing becomes democratized and gut metagenomes are explored in understudied populations such as those in Southeast Asia, the strategy and value for establishing population-specific MAG references remains an open question.

Leveraging the availability of a healthy Singaporean adult cohort comprising three ethnicities (Chinese, Malay and Indian), two of which (Chinese and Malay) represent significant populations in Southeast Asia, we deeply characterized 109 gut metagenomes with state-of-the-art hybrid sequencing (short and long read) and Hi-C technologies (*Singapore Platinum Metagenomes Project* – SPMP). The resulting datasets were assembled to produce high-quality references that significantly improve existing databases in assembly quality (>28× N50 improvement), helped identify 70 previously uncharacterized gut microbial species (>10% novel) and more than 3400 strains in Southeast Asian populations, and uncovered thousands of novel BGCs that serve as a resource for bioprospecting. The ability to substantially augment existing databases through in-depth hybrid metagenomic analysis highlights the value of this strategy applied to understudied geographical regions like Southeast Asia, the importance of uncharacterized Asian microbial diversity, and serves as a template for population-specific 'platinum' metagenome references for precision medicine programs around the world.

## Results

### Generation of a population-specific high quality gut microbial reference catalog

To explore the utility of various metagenomic strategies for generating a high-quality gut microbial reference database for a population, subjects from an existing multi-omics study in Singapore[27] were recruited to provide stool samples with informed consent (n = 109; Supplementary Data 1, "Methods"). Samples were collected using a kit designed for preserving anaerobes, DNA was extracted with a protocol optimized for high molecular weight, and shotgun sequencing was performed using short (Illumina, 2 × 151 bp, average depth = 9.4Gbp, Supplementary Data 2) and long read (Oxford Nanopore Technologies - ONT, median N50 = 8.6kbp, average depth = 5.8Gbp, Supplementary Data 2) technologies, along with high-throughput chromosome conformation capture (Hi-C) analysis for a subset of samples (n = 24; Supplementary Fig. 1, Supplementary Data 2, "Methods"). The distribution of taxa in both sequencing technologies (Illumina and ONT) were confirmed to be highly concordant (median correlation coefficient $\rho$ = 0.90), enabling joint analysis of both datasets (Supplementary Fig. 2).

We next compared the commonly used short-read strategy for building MAG reference collections[13,20–22], with a recently proposed hybrid assembly strategy[25], for their utility in building a population-specific database ("Methods"). From a cost perspective, we noted that the hybrid strategy required <$150 in additional sequencing costs per sample (~100% increase in total cost) and marginal increase in cloud computing cost per sample ("Methods"). This in turn was observed to result in >61% increase in the number of genomes produced per sample (>15 additional MAGs; Fig. 1a) with the hybrid strategy, with some samples yielding >80 genomes. Overall, 4497 MAGs were obtained with hybrid assembly for 109 samples, versus 2789 MAGs with short-reads alone (Supplementary Data 3), with several abundant gut bacterial genera having enhanced representation within hybrid assemblies (e.g., *Bifidobacterium*, *Faecalibacterium* and *Blautia*; Fig. 1b). This was observed to substantially improve read assignment to the reference genome database, ensuring that more genomes were detected (n = 217 vs n = 119), and with computed relative abundances being more consistent with kraken abundances for hybrid assemblies versus short-read assemblies (median relative error = 8% vs 73%; Fig. 1c, Supplementary Fig. 3). Overall, hybrid assemblies consistently improved the recovery of genomes across genera, with no significant bias to any specific genera, highlighting the versatility of this approach (Supplementary Fig. 4).

Incorporation of long-read data in hybrid assemblies enabled marked improvements in assembly contiguity (>28×) as reported previously[25], with an average N50 of 339kbp (L50 = 12) with hybrid assembly relative to an N50 of 12 kbp with short reads alone (Fig. 1d). This was also accompanied by a notably lower level of chimerism (<10% vs >20% with short-read assemblies) and similar annotated gene lengths as short-read assemblies (Supplementary Fig. 5), suggesting that hybrid assemblies are robust to indel errors in long reads. Overall, this provided higher quality genomes based on MIMAG critera[28] after binning[10], where many hybrid MAGs had correctly reconstructed rRNA genes[29], and no such MAGs were obtained with short-read only assembly (Fig. 1e, "Methods"). To assess if the quality of MAGs could be improved further, Hi-C data was used to assist in contig binning[23,30–34]. This was found to marginally increase the proportion of high-quality MAGs obtained, and double the proportion of near-complete genomes, with similar average assembly contiguity (Supplementary Fig. 6 and Supplementary Data 3). As the per sample cost of Hi-C analysis is currently high (>$500), studies for generating population-specific references will need to consider this cost-benefit tradeoff.

Hybrid assembled genomes in SPMP were assigned taxonomy based on the Genome Taxonomy Database[2] (GTDB) and compared to GTDB reference genomes to assess their utility. SPMP genomes were found to provide notably improved references for most GTDB species, for both isolates (>6× increase in N50) as well as uncultivated organisms (>13×; Fig. 1f). While the improvement in assembly is expected for uncultivated organisms that are primarily assembled using short-read metagenomics, the observed improvement relative to GTDB isolates (albeit smaller, Wilcoxon $p$-value = $1.25 \times 10^{-11}$) is noteworthy as assembling the latter is typically less error prone. Overall, SPMP genomes provided high-quality references for 110 GTDB species, 46 of which have isolates, highlighting the value of a 'platinum' metagenomics approach for augmenting existing reference genome databases (Fig. 1g and Supplementary Data 4).

### Asian gut microbiomes harbor substantial uncharacterized gut microbial genetic diversity

By encompassing three major Asian ethnicities (Chinese, Malay, Indian) in Singapore we anticipated that the SPMP would be a useful resource to explore Southeast Asian gut microbial diversity and tested the idea of population-specific MAG reference catalogs (Supplementary Fig. 7). Subsampling based rarefaction analysis with SPMP MAGs showed that with as few as 100 subjects, >90% of the estimated recoverable (at the genomic level) gut microbial species diversity of the Singaporean population was represented in the SPMP catalog (Fig. 2a, "Methods"). Similarly, with a genome collection that is 1/6th the size of the species-level public gut microbial reference database[13] (UHGG; 18Gb vs 3Gb), the strain-level SPMP database can be used to identify more gut bacterial reads from an independent Singaporean study[35] (92% vs 91%), and classify substantially more reads at the genome-level when database sizes are similar (83% vs 66%; Supplementary Fig. 8). Furthermore, the

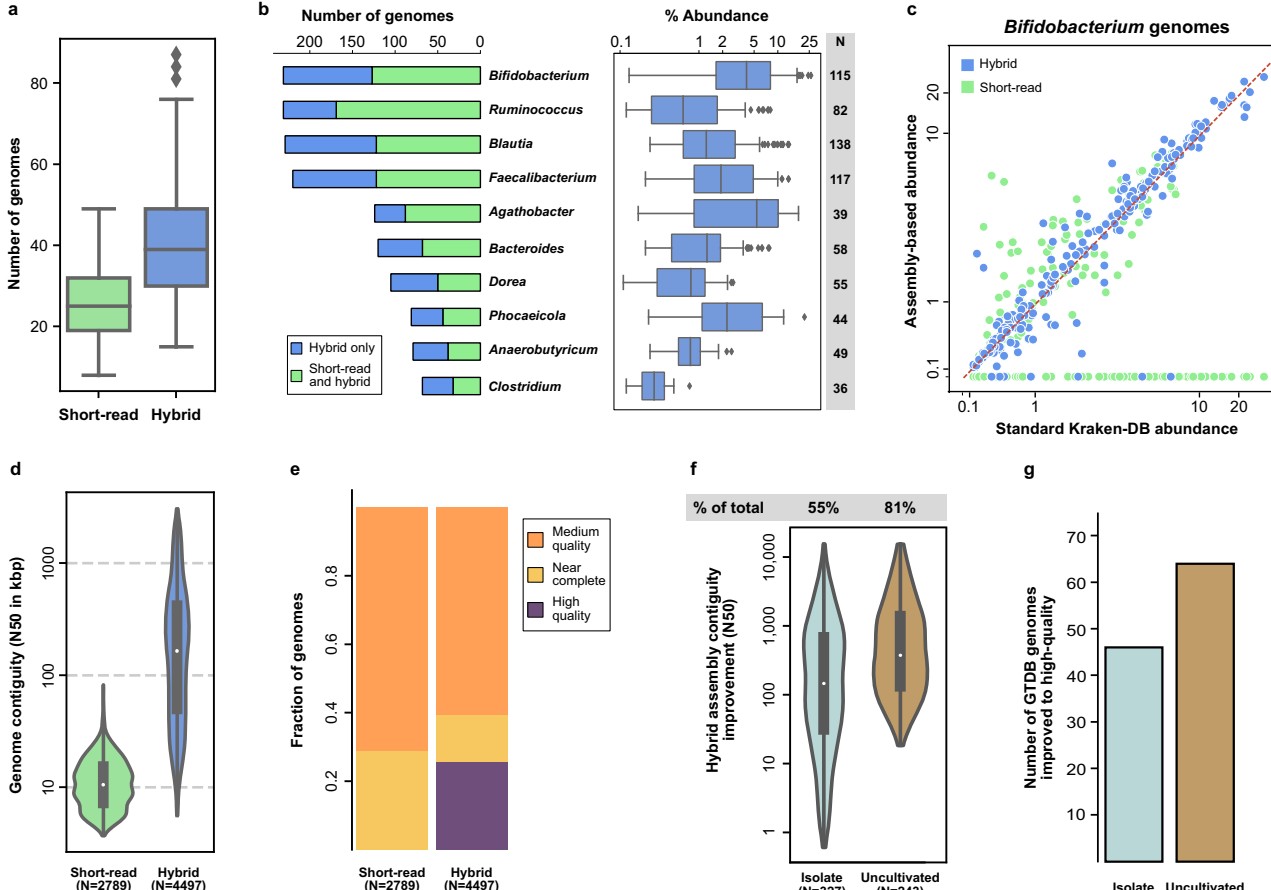

**Fig. 1 | Assembly strategy for high-quality microbiome references. a** Boxplots showing the number of MAGs obtained across metagenomic datasets using short-read and hybrid assemblies (n = 109). **b** Stacked barchart showing genus-specific breakdown of the number of MAGs obtained using short-read and hybrid assemblies (left) and boxplots for corresponding relative abundances of the genera (right). N represents the number of hybrid only MAGs for each genus. **c** Scatter-plot showing the relative abundance of *Bifidobacterium* genomes estimated using short-read or hybrid assemblies for a sample (*y*-axis) versus corresponding relative abundances obtained using the standard Kraken2 database (*x*-axis). Points found along the *x*-axis represent *Bifidobacterium* species found using the Kraken standard database but not found using either short-read or hybrid MAGs. **d** Violin plots showing the distribution of a contiguity metric

(N50 – largest contig size where >50% of the genome is in larger contigs) for short-read and hybrid assembly based MAGs. **e** Stacked barcharts showing the relative proportion of MAGs satisfying different MIMAG quality standards with short-read and hybrid assemblies of SPMP datasets. **f** Violin plots showing the relative improvement in contiguity (N50) obtained using hybrid assembly MAGs from SPMP relative to matched genomes in the GTDB database. **g** Barcharts showing the number of GTDB reference genomes which were improved from medium to high MIMAG quality using SPMP MAGs. Center lines in the boxplots represent median values, box limits represent upper and lower quartile values, whiskers represent 1.5 times the interquartile range above the upper quartile and below the lower quartile, and all data points are represented as dots in the figures. Source data are provided as a source data file.

SPMP database provided higher sensitivity for strain-level read mapping relative to UHGG even when the number of strains available were matched across species ("Methods"). These results indicate that a high-quality database with population-specific strain-level representation can provide better references for microbiome read mapping or taxonomic classification, while potentially using fewer computational resources, by capturing both relevant species and strain genomes found in the corresponding populations.

To understand microbiome variability across ethnicities and its utility to discover new biological insights, we used multivariate regression analysis[36] to explore relationships between gut metagenome composition and demographic factors (e.g., sex, age, and ethnicity). Interestingly, more than 60% of the taxonomic associations discovered (91 out of 133; MaAsLin2 *p*-value<0.05) were related to ethnicity, with 23 gender-specific and 19 age-based associations (Supplementary Data 5). We then aggregated SPMP MAGs into species-level clusters (SLCs, 95% identity), annotating them with publicly available reference genome collections (Supplementary Fig. 9, "Methods") to identify 70 putative new species for which no genomes have been available previously, despite large-scale MAG generation efforts[2,13] (Fig. 2b). Surprisingly,

these putative new species represent >10% of the SLCs obtained (n = 685) and are in addition to the 363 clusters that only have MAGs and no isolate genomes in existing databases (GTDB: https://gtdb.ecogenomic.org/, based on systematic analysis of curated genomes in RefSeq: https://www.ncbi.nlm.nih.gov/refseq/ and GenBank: https://www.ncbi.nlm.nih.gov/genbank/). More than 50% of the novel SLCs (38 out of 70) were only assembled with hybrid assembly and were missing in short-read assemblies. In addition, hybrid assemblies provided a >13× median N50 improvement overall, generating nearly all of the high-quality and near-complete genomes for the novel SLCs (19 out of 20), highlighting the utility of this strategy for capturing microbial diversity. In comparison to a recently published resource for under-represented East and South Asian populations[22] we found that most species were still novel (87%, 61/70) emphasizing the importance of generating population-specific references.

Among the novel SLCs, in addition to representatives in nearly all orders commonly containing gut microbes (e.g., Bacteroidales), we noted that 17 could be classified to the order Coriobacteriales while an additional 7 were assigned to Christensenellales, both of which are relatively understudied gut bacterial orders with high diversity in

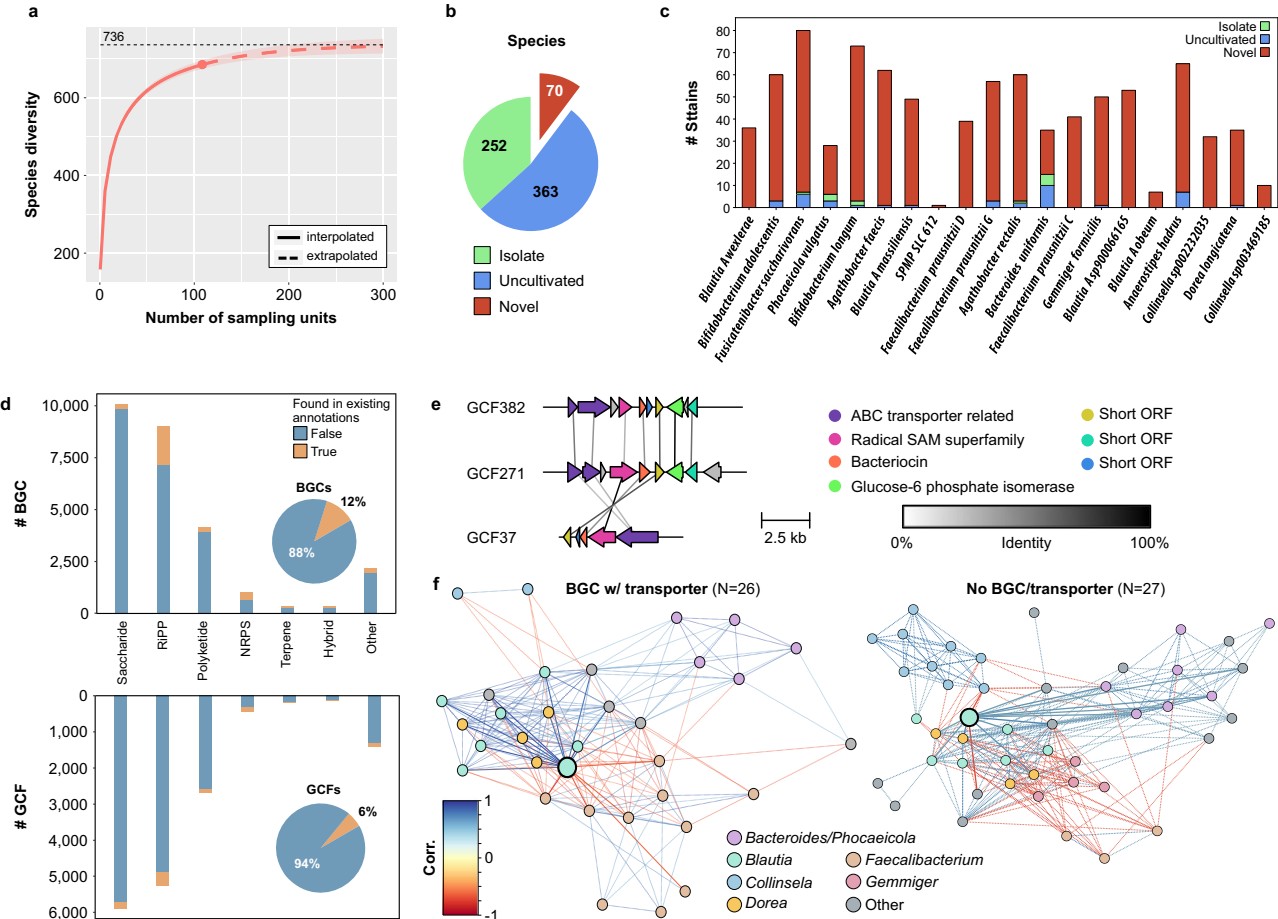

**Fig. 2 | Characterization of novel species, strains and gene families in SPMP genomes. a** Rarefaction analysis showing that the SPMP database covers a substantial fraction of the species level diversity in its MAGs. Error bands represent confidence intervals of 95%. **b** Pie-chart showing the breakdown of species-level clusters in SPMP that have an *isolate* genome, only have MAGs (*uncultivated*) and are *novel* compared to genomes in public databases (UHGG, GTDB, SGB). **c** Stacked barcharts showing the number of SPMP strains that have an *isolate* genome, only have MAGs (*uncultivated*), and are *novel* compared to all UHGG genomes (>200,000, <99% ANI). The species shown are the top 20 in terms of median relative abundance in SPMP (most abundant on the left). **d** Stacked barcharts showing the number of BGCs (top) and GCFs (bottom) in different product classes that are present or absent in existing annotations comprising of the antiSMASH and MiBIG databases as well as antiSMASH annotations from HRGM. Inset piecharts show the overall breakdown. **e** Synteny plots showing the conservation of gene order and orientation (colored arrows, relatedness shown by vertical lines) for a novel GCF (GCF382) and related families. **f** Network diagrams depicting correlations between gut microbial species (nodes – species, edges – significant correlations) and overall microbiome structure in SPMP metagenomes when stratified based on presence or absence of GCF 382/271/37 (or missing the corresponding transporter gene) in a *Blautia* species (enlarged teal node, solid edges to correlated species, dashed edges between other nodes). Source data are provided as a source data file.

general and few isolates (Supplementary Fig. 10). Additionally, three novel SLCs with high-quality MAGs represent the only available genomes for the corresponding genera (SLC637 – closest match *Phocaeicola*, <83% identity; SLC487 and SLC667 – closest match *Butyricicoccus*, <81% identity), while one of the novel SLCs is among the top 10 most abundant SLCs within the gut microbiomes of SPMP subjects (SLC612; Supplementary Fig. 11). We noted that SLC612 is significantly more abundant in the gut microbiomes of Singaporean populations than in western subjects, potentially explaining why it was not assembled in previous large-scale studies and emphasizing the need for population-specific references for even common gut bacteria (Supplementary Fig. 11).

At the strain-level (99% identity), SPMP genomes were notably unique compared to >200,000 genomes in the UHGG database, with 3413 novel strains out of 3891 (87% novel, Methods). Among the top 20 most abundant gut bacterial species in SPMP, less than 20% of the strains were represented in UHGG, with only the keystone gut commensal *Bacteroides uniformis* having >40% of its strains being represented by genomes from other populations (Fig. 2c). For species that are extensively characterized due to their use as probiotics such as

*Bifidobacterium adolescentis* and *Bifidobacterium longum*, we noted that while many strain genomes have been obtained from isolates (>30; Supplementary Fig. 12), SPMP MAGs reveal an even greater uncharacterized diversity in the Singaporean population (>50 novel strains; Fig. 2c, Supplementary Fig. 12) that could be leveraged for probiotic discovery.

To explore the utility of the SPMP database for bioprospecting and discovering secondary metabolic pathways that may be important for gut microbiome structure and function, we combined comparative[37] and deep learning[38] based approaches for annotating biosynthetic gene clusters with high stringency filters (BGCs, "Methods")). In total, we identified 27,084 BGCs (DeepBGC: 23,175; antiSMASH: 3909) that grouped into 16,055 gene cluster families by BiG-SCAPE[39] (GCFs; Fig. 2d). More than 90% of the GCFs (15,134) did not display similarity to previously known BGCs in curated standard databases (antiSMASH and MIBiG) and were not found in annotations within an extensive collection of gut microbial reference genomes (HRGM, "Methods"), highlighting the value of using complementary algorithms for bioprospecting in new populations. We estimated that >85% of SPMP GCFs were not represented in curated databases, even

when only a higher confidence set of predictions from antiSMASH was considered, while 49% of GCFs were novel even after taking into account more extensive HRGM antiSMASH annotations (Supplementary Figs. 13 and 14).

While a significant fraction of GCFs were predicted to encode for saccharides ($N$ = 5888, 37%), in line with their important functions in microbe-microbe and microbe-host interactions[40], many novel GCFs appear to encode diverse bioactive compounds such as ribosomally translated and post translationally modified peptides (RiPPs), polyketides and non-ribosomal peptides (NRPs) (Fig. 2d), some of which may have antimicrobial function ("Methods"). In particular, a group of GCFs not represented in curated databases was predicted to synthesize a bacteriocin in a *Blautia* species, with 3 distinct gene configurations and genes encoding enzymes for peptide modification (radical SAM superfamily) and ABC transporter genes (GCF382/271/37, Fig. 2e). Analyzing the structure of the microbial community in samples with and without the novel GCFs identified distinct networks, with presence of GCF382/271/37 associated with strong negative correlations between the *Blautia* species and multiple *Faecalibacterium* species including *Faecalibacterium prausnitzii* (Fig. 2f, "Methods"). Together with the known role of *Faecalibacterium* species in gut health[41,42], these observations highlight the importance of comprehensively identifying secondary metabolic pathways for understanding gut metagenome function in human diseases.

## Discussion

Despite the growing number of gut microbiome studies worldwide, including from remote populations in the Americas[43] and hunter-gatherer tribes in Africa[44], the gut microbial diversity of Asian populations remains understudied[45]. Singapore represents a microcosm of multiple major Asian ethnic populations (Chinese, Malay and Indian) living in the shared environment of a modern metropolis. While there has been extensive study of gut metagenomes of ethnic Chinese individuals from China, fewer studies have involved individuals from Southeast Asia and India. The SPMP can thus represent an important reference for these populations, in addition to Singaporean studies. We believe that SPMP is only the beginning of such efforts because our data is unlikely to represent the entirety of microbial diversity even in Southeast Asia alone. More broadly, we anticipate that the microbial diversity seen in SPMP might be similar to what would be observed in other major urban centers in Asia (e.g., New Delhi, Jakarta, Tokyo, Hong Kong), but is likely the 'tip of the iceberg' when considering rural and nomadic populations.

Various parameters are likely to define the appropriate strategy for a study similar to SPMP in other countries, including cost, targeted quality of reference genomes, ease of technology access, and availability of sufficient number of samples from a representative baseline cohort in the country. While we attempted to employ multiple different technologies for SPMP to get high-quality assemblies, we chose the middle-ground in terms of cost and accessibility as this is an important consideration for many countries. In particular, even higher-quality metagenomic assemblies are possible if HiFi reads from the Pacific Biosciences Sequel IIe system are available[46]. Also, the recent announcement of higher-quality reads from ONT could help improve assembly further and reduce costs[47]. Even as the sequencing landscape is constantly changing, the results from our study suggest that high-quality population-specific metagenomic references are already feasible with a modest-sized cohort and limited sequencing resources.

The advantages of having high-quality references for metagenomics are similar to what other areas of genetics and studies in model organisms have benefited from i.e., substantially reduced cost and effort in future studies by: (i) allowing the use of short reads or a single sequencing assay/technology, (ii) enabling increased sensitivity in identification of genomic features using reference-based approaches (e.g., taxonomic classifiers for metagenomics), (iii) ensuring that there

are fewer 'dark matter' reads whose origin is unknown. We envisage that efforts such as SPMP will benefit the scientific community by spurring greater adoption of reference-based analyses in metagenome-wide association studies[48,49]. Additionally, as we noted in Fig. 1f, g, the quality of genomes (measured by higher contiguity and completeness) that can be obtained using metagenomics and hybrid assembly can be comparable or better than genomes of microbial isolates hosted on GTDB, of which a substantial proportion were sequenced using short reads only. Furthermore, unlike MAGs assembled from short reads only, SPMP MAGs have low levels of contamination and chimerism on average, increasing their utility as population-specific references. This can galvanize efforts to genetically map microbial ecosystems in diverse biospheres, further contributing to the references available to study human microbiomes and understanding of strain sharing between humans and the environment. As sequencing costs, ease of use and accessibility of new technologies, and metagenomic assembly algorithms improve, we can expect that a majority of the high-quality microbial references that will be used in the future would be obtained through metagenomics, thus helping to bridge the knowledge gap for the hundreds of thousands of microbial species that are estimated to be there on Earth.

The detection of 70 putative novel species in SPMP is perhaps not surprising given the unexplored microbial diversity and the limitations of current genetic databases. However, it is noteworthy that this is still a substantial fraction of the species detected in this study (>10%, Fig. 2b), and while some of these species are not frequently detected across individuals, one of them was in the top 10 most abundant gut bacterial species, while others may still play a significant role in the biology of some individuals by being sporadically abundant (e.g., SLC665 which is among the top 20 most abundant species in 5% of subjects). Not surprisingly, at the strain-level an even larger fraction of the observed genetic diversity was novel, but what was notable was that this was true even for the more abundant and well-studied species in the gut microbiome (e.g., *Bacteroides uniformis* and *Bifidobacterium adolescentis*, Fig. 2c). These observations highlight the overall value of such studies for discovering probiotic strains that could be leveraged for population health, with modest investments in metagenomic analysis cost (<$40,000), making it feasible for national microbiome projects around the world.

Finally, the identification of >23,000 BGCs in the SPMP database that were not represented in existing annotated databases (88% of total, Fig. 2d) highlights that we are only scratching the surface in terms of harnessing microbial pathways and functions for synthetic biology and biotechnology applications. This was made possible by the high-contiguity of our hybrid assemblies (>28× N50 relative to short-read assemblies), and the characterization of distinct, under-represented South-East Asian populations in SPMP harboring substantial novelty relative to curated BGC databases (>85%) and annotated reference genomes (49%, Supplementary Figs. 13 and 14). The gut microbiome by virtue of being a dynamic, host-associated community with high diversity of microbes is a rich hunting ground for host-modulating, macro-nutrient catabolizing and micro-nutrient synthesizing functions[50,51]. In addition, homeostasis in the gut microbiome may be maintained by key members of the community through the selective expression of antimicrobial peptides[52] (AMPs), and correspondingly we identified hundreds of novel BGCs encoding putative bacteriocins, sactipeptides, lanthipeptides and lassopeptides that can now be further characterized ("Methods"). Notably, we found evidence that the presence of a BGC in a common *Blautia* species is associated with significant changes in overall gut microbiome community structure for SPMP subjects (Fig. 2f). Together these results highlight the potential for novel AMPs discovered in SPMP to provide genetic templates for further optimization, and subsequent use to modulate the gut microbiome, or as new antimicrobials to target multi-drug resistant pathogens.

## Methods

### Subject recruitment

Subjects for this cross-sectional study were recruited based on recall from a community-based multi-ethnic prospective cohort[27] that is part of the Singapore Population Health Studies project (SPHS -formerly Singapore Consortium of Cohort Studies). This subset included 109 subjects who were 48 to 76 years old with 65 males and 44 females (Supplementary Data 1). Subjects in SPHS were recruited to participate in the National Health Survey, where subjects were selected at random using age- and gender- stratified sampling to obtain a representative sample set of residents in the country. During the period of recruitment from April 16th, 2008 to September 20th, 2018, subjects did not have any pre-existing major health conditions (cardiovascular disease, mental illness, diabetes, stroke, renal failure, hypertension and cancer) based on self-reporting[27]. The ethnicity of each subject was confirmed verbally so that all four grandparents of the subject belonged to the same ethnic group. As such, we do not anticipate that any self-selection bias was introduced. A separate comparison of baseline clinical measurements was performed, including age-adjusted BMI and HbA1c, against the rest of the subjects in the larger ethnicity-specific cohorts within Singapore Population Health Studies to ensure that the sampling for the initial cohort conformed to population norms. Informed consent was obtained from all participants. Each subject was given 60 Singapore Dollars for their participation in this study. All associated protocols for this study were approved by the National University of Singapore Institutional Review Board (IRB reference number H-17-026) on May 9th, 2017 and renewed until May 31st, 2021.

### Sample collection

Fecal samples were collected from healthy subjects using the Bio-Collector™ kit (The BioCollective, Colorado, USA). Samples were double-bagged and transferred to a polystyrene box, together with a pre-chilled ice-pack (−20 °C). The polystyrene box was transferred to a cardboard box and later collected from the participants' home within the same day. Samples brought to the Temasek Life Sciences laboratory were stored into an anaerobic chamber (atmosphere of $N_2$ (75%), $CO_2$ (20%), and $H_2$ (5%)). Fecal samples were homogenized and sub-samples transferred into sterile 2 mL centrifuge tubes.

### DNA extraction

Genomic DNA was extracted from fecal material (0.25 g wet weight) using the QIAamp Power Fecal Pro DNA kit (QIAGEN GmbH, Cat. No. 51804) and was quantified using Qubit dsDNA BR Assay Kit (Thermo Fisher Scientific, Cat. No. Q32853). Integrity of the extracted DNA was verified using 0.5% agarose gel electrophoresis.

### Illumina library preparation and sequencing

Metagenomic libraries were prepared with a standard DNA input of 50 ng across all samples, using NEBNext® Ultra™ II FS DNA Library Prep Kit for Illumina (New England Biolabs, Cat. No. E7805), according to the manufacturer's instructions. The reaction volumes were, however, scaled to a quarter of the recommended volumes for cost effectiveness. Barcoding and enrichment of libraries was carried out using NEBNext® Multiplex Oligos for Illumina® (96 Unique Dual Index Primer Pairs; New England Biolabs, Cat. No. E6440). Paired-end sequencing (2 × 151 bp reads) was carried out on the Illumina HiSeq4K platform with a minimum and average depth per sample of 2.4 Gb and 9.4Gb respectively.

### ONT library preparation and sequencing

Purity and integrity of DNA was assessed and ensured to fall within recommended ranges before library preparation. To preserve the integrity of DNA, the shearing step was omitted and DNA was used directly for DNA repair and end-prep. Single-plex libraries were prepared using 1D sequencing kit (Oxford Nanopore Technologies, SQK-

LSK108 or SQK-LSK109) according to the "1D Genomic DNA by ligation" protocol. For samples that were multiplexed (12-plex), the native barcoding kit (Oxford Nanopore Technologies, EXP-NBD103 or EXP-NBD104 and EXP-NBD114) was used and libraries were prepared according to the "Native barcoding genomic DNA" protocol. Both native barcode ligation and adapter ligation steps were extended to 30 min instead of 10 min. Single-plex samples were sequenced on either the MinION or GridION machine with either FLO-MIN106D or MIN106 revD flowcells. Multiplex samples were sequenced on the PromethION machine with FLO-PRO002 flowcells. Raw reads were basecalled with the latest version of the basecaller available at the point of sequencing (Guppy v3.0.4 to v3.2.6). Basecalled nanopore reads were demultiplexed and filtered for adapters with qcat (v1.1.0 https://github.com/nanoporetech/qcat). The minimum and average sample depth was 1.2 and 4.7Gb respectively. Number of reads ranged from 300,000 to 3.4 million (average = 1.4 million).

### Hi-C library preparation and sequencing

Hi-C libraries were generated using Phase Genomics ProxiMeta kit (version 3.0), based on the standard protocol. Briefly, 500 mg fecal material was crosslinked for 15 min at room temperature with end-over-end mixing in 1 mL of ProxiMeta crosslinking solution. Once crosslinking reaction was terminated, quenched fecal material was rinsed. Sample was resuspended and a low-speed spin was used to clear large debris. Chromatin was bound to SPRI beads and incubated for 1 h with 150 μL of ProxiMeta fragmentation buffer and 11 μL of ProxiMeta fragmentation enzyme. Once washed, beads were resuspended with 100 μL of ProxiMeta Ligation Buffer supplemented with 5 μL of Proximity ligation enzyme and incubated for 4 h. After reversing crosslinks, the free DNA was purified with SPRI beads and Hi-C junctions were bound to streptavidin beads and washed to remove unbound DNA. Washed beads were used to prepare paired-end deep sequencing libraries using ProxiMeta Library preparation reagents. Paired-end sequencing (2 × 151 bp reads) was carried out on the Illumina HiSeq4K platform. The minimum and average sample depth was 2.3 and 24.5 Gb respectively.

### Estimating sequencing and computing cost

Sequencing costs can vary substantially across sequencing centers and countries. Here we provide an estimate based on costs at the Genome Institute of Singapore in November 2021. Based on prices for library preparation kits as described in this manuscript, we estimate that Illumina library preparation costs ~US$50/sample and an Illumina HiSeq sequencing lane costs ~US$1000 with approximate throughput of >350 million paired-end reads (2 × 151bp; >100Gbp). Considering that the average Illumina sequencing depth per sample in this study is ~10Gb, 10 samples can be multiplexed in a single lane, leading to the overall cost per sample of ~US$150. For ONT sequencing, we estimate that with an approximate flow-cell price of US$500 producing ~30Gbp of sequencing data, 5 samples can be multiplexed to obtain the average throughput in this study (~6Gbp). With ONT multiplexed library preparation costs of ~US$50/sample, we estimate that overall ONT costs are also ~US$150/sample. Metagenomic assembly of Illumina and Hybrid datasets with MEGAHIT and OPERA-MS, respectively, typically took less than 3 h on an AWS C5 instance with 8 CPUs. Using as reference an AWS C5 instance price of 30 cents an hour for 8 CPUs, this translated to a computational cost of <US$1/sample on average, a marginal increase over total sequencing costs.

### Sequence quality assessment

Illumina and ONT read statistics were generated with Fastq-Scan (v0.4.1, https://github.com/rpetit3/fastq-scan) and NanoStat[53] (v1.4.0), respectively. To assess taxonomic concordance, Illumina and ONT reads were classified with Kraken2[54] (v2.1.1, UHGG database[13]) and relative abundances were estimated with Bracken[55] (v2.6.1) at the

species level (option -l R7) to compute Pearson correlation coefficients per sample.

## Metagenomic assembly and binning

Illumina reads were assembled using MEGAHIT[8] (v1.04, default parameters) and hybrid metagenomic assemblies were generated with Illumina and ONT data using OPERA-MS[25] (v0.9.0, --polish). Contigs were binned with MetaBAT2[10] (v2.12.1, default parameters). Hi-C binning was provided by Phase Genomics using its internal pipeline with MetaBAT results for hybrid assemblies as a starting point. Assembly bins were evaluated based on MIMAG standards[28], with contamination, completeness and N50 values determined with CheckM[56] (v1.04), and non-coding RNA annotations from barrnap (https://github.com/tseemann/barrnap) (v0.9) and tRNAscan-SE[57] (v2.0.5, default parameters). Assembly bins with contamination <10% and completeness >50% were designated as *medium quality* MAGs, those with contamination <5% and completeness >90% as *near complete* MAGs, and additionally near complete MAGs with complete 5S, 16S, and 23S rRNA genes and at least 18 unique tRNA genes were classified as *high quality* MAGs. All other bins were classified as *low quality* and were removed from further analyses. In total, 4497 medium quality, near complete and high quality MAGs were designated as being part of the SPMP database. Hybrid and short-reads assembly based MAGs were further assessed for chimerism with GUNC[58] (v1.0.4, detailed output). Coding sequence lengths obtained from Prodigal[59] (v2.6.3) calls were compared between the two datasets to assess the potential impact of long read indel errors on gene annotation. Concordant with prior work showing that hybrid metagenomic assemblies can have high base-pair accuracy[25], we also noted that SPMP MAGs independently assembled from distinct individual gut metagenomes could exhibit high average nucleotide identity (>99.99%, consistent with Q40 quality).

## Species abundance and rarefaction analysis

Representative MAGs for SLCs were used to create a custom Kraken[60] (v2.1.1) database (https://github.com/DerrickWood/kraken2/wiki/Manual#custom-databases) and relative abundances for SLCs were estimated for each sample using Bracken[55] (v2.6.0, default parameters). Rarefaction analysis for estimating overall species diversity was done using the R package iNext[61] (v2.1.7, q = 0, datatype = "incidence_raw" and endpoint=300), based on converting SLC relative abundance values from Bracken into presence-absence values at a threshold of 0.05%.

## Multivariate regression analysis

Genus-level abundances for each sample were provided as input for R package MaasLin2[36] (v1.4.0) along with sample metadata (age, sex and ethnicity), and significant associations were determined by combining 3 MaasLin2 runs with a compound Poisson linear model.

## Strain-level read mapping

Metagenomic reads were mapped (--secondary=no) against reference databases indexed with minimap2[62] (v2.24-r1122, -I 24 G; SPMP strain-level genomes and UHGG species-level representatives). Alignments were filtered at the strain-level with bamtools (v2.5.2, -tag "NM: < 2" -length ">99") and unique reads were extracted based on samtools (v1.15.1) view results.

To further evaluate the utility of SPMP genomes relative to the UHGG database for read mapping at the strain-level, we created databases with similar number of strains from both collections. Reference indexing and mapping were done in a similar fashion as described before. Alignments were filtered with pysam (v0.19.1) (read coverage ≥90%, identity ≥99%), and reads were classified at the species-level with Kraken (v2.1.1, RefSeq bacteria database). Specifically, we identified 21 species with many strain genomes in UHGG or SPMP (≥20) and having enough reads (>10× coverage) in at least

3 samples in an independent study of Singaporean gut metagenomes[35]. Illumina reads were mapped (minimap2, default parameters) independently to strain genomes for each species. Kraken2 classification (standard database) was used to assess if mapped reads came from the right species, and to calculate precision, sensitivity and F1 scores. We noted that median F1 scores were better using SPMP compared to UHGG for 17 out of 21 species. Overall, SPMP provided significantly better mapping performance (F1 score) relative to UHGG for 12 species (Wilcoxon p < 0.05). The converse, i.e., significant improvements with UHGG relative to SPMP, were not observed for any species. Improvements in F1 scores were driven by better sensitivity in SPMP vs UHGG for abundant gut bacterial species such *Prevotella copri* and *Alistipes onderdonkii*. While median precision scores using SPMP and UHGG were similar (0.98 vs 0.99 for *P. copri*; 0.98 vs 0.97 for *A. onderdonkii*), sensitivity was notably higher in SPMP vs UHGG (0.96 vs 0.90 for *P. copri*; 0.99 vs 0.90 for *A. onderdonkii*).

## Annotation of MAGs with the Genome Taxonomy Database

The SPMP database was compared to the GTDB database[2] (release 95) using GTDBtk's[63] (v1.4.1) ani_rep command with default arguments, which leverages Mash[64] (v2.3) to provide pairwise genome-wide similarity values between all query MAGs and GTDB sequences. Only pairs with Mash distance ≤0.05 were retained and used to define the best match for each SPMP MAG based on minimum Mash distance. GTDB matches were classified based on their metadata as being *uncultivated* ("derived from environmental sample" or "derived from metagenome") or based on *isolate* strains. Both N50 values and MIMAG classifications were extracted from GTDB metadata. MAGs were placed into a phylogenetic tree using GTDB_TK (v1.4.1) with classify_wf (default options), based on pplacer_taxonomy values. To assess novelty in light of the latest human gut metagenome database, we further compared our MAGs to the 5414 representative genomes from the Human Reference Gut Microbiome catalog (HRGM)[22] with a similar Mash analysis (Supplementary Data 6).

## Species and strain-level clustering

MAGs were clustered at the species (95%) and strain-level (99%) based on average nucleotide identity estimates (ANI; using Mash with sketch size of 10k and k-mer size of 21 bp) with agglomerative clustering (sklearn v0.23.2, AgglomerativeClustering function, options: linkage = "single", n_clusters=None, compute_full_tree=True, affinity = "precomputed"). For each cluster, *representative* MAGs were defined using the highest eigen centrality value based on a weighted network graph produced by networkx (v2.5; eigenvector_centrality function). Strain-level clustering was done jointly with all species-level matches from the UHGG database (v1.0, ANI threshold of 95%). Phylogenetic analysis at the strain-level was conducted using the biopython Phylo package[65], based on pairwise distances generated with FastANI[66] (v1.32). Phylogenetic trees were visualized using FigTree (tree.bio.ed.ac.uk/software/figtree).

## Species assignment

SLCs were assigned putative species name and types based on comparisons with multiple databases, including GTDB, Pasolli et al.[67] (SGB) and Almeida et al.[13] (UHGG). SLCs types were defined as, (i) isolate: if GTDB match to an isolate was found (Mash distance ≤0.05), (ii) uncultivated: if a match to any database was found, but no isolates, (iii) novel: if no matches were found. SLCs were assigned putative species names based on a majority rule for MAGs in the cluster, with preference for GTDB ids (Supplementary Fig. 9).

## Biosynthetic gene cluster identification and clustering

Biosynthetic gene clusters (BGCs) in the SPMP database were identified using antiSMASH[68] (v5.1.2, --genefinding-tool prodigal-m --cb-general --cb-knownclusters --cb-subclusters --asf --pfam2go --smcog-

trees) and DeepBGC[38] (v0.1.18, prodigal-meta-mode). BGCs with only one identified gene and with length <2kbp were removed for both sets of results. For antiSMASH this provided a set of 3,909 BGCs. DeepBGC results which overlapped with antiSMASH were removed if the genomic coordinates of both BGCs overlapped by ≥30% in either direction. DeepBGC candidates were further filtered for (i) being categorized with a known product class and (ii) containing at least one known biosynthetic pfam or TIGRFAM protein domain as defined by Cimermancic et al.[69], providing an additional set of 23,175 BGCs.

All 27,084 BGCs (3909 from antiSMASH + 23,175 from DeepBGC) were first categorized into different product classes: ribosomally synthesized and post-translationally modified peptides (RiPPs), non-ribosomal peptide synthetases (NRPs), polyketide synthases (PKS), saccharides and others based on the labels reported by each algorithm. We further unified the antiSMASH and DeepBGC product class labels to integrate both datasets (Supplementary Table 1). A fraction of mined BGCs were labeled as "hybrids" because antiSMASH or DeepBGC associated them with two different product classes e.g., "bacteriocin;T1PKS". The BGCs in each product class were grouped into gene cluster families (GCFs) by sequence similarity using BiG-SCAPE[39] (v1.01, --include_singletons --mix --no_classify --cutoffs 0.3). A total of 16,055 GCFs were defined by this approach and for each GCF we took the smallest BGC member as a representative of the family. Gene cluster diagrams of BGCs were created using Clinker[70].

BGCs in SPMP were classified as *novel* via a two-step approach. Firstly, BGC sequences were required to have <80% similarity to any existing sequence in the antiSMASH and MIBiG 2.0[71] databases using the clusterblast results from antiSMASH. Secondly, BGC annotations were compared to antiSMASH annotations from a comprehensive gut microbial genome collection (HRGM) using the standalone clusterblast software[72] (v 1.1.0), to identify SPMP matches based on a 80% similarity threshold, similar to the approach described in Gallagher et al[73].

### Characterization of AMPs and impact on microbiome structure

Besides bacteriocins, BGC mining in the SPMP database also identified other classes of ribosomally synthesized and post-translationally modified peptides (RiPPs) such as lanthipeptides and lassopeptides (Supplementary Fig. 15A), which can also possess antimicrobial properties. Antimicrobial activities of putative peptides encoded by novel RiPP BGCs in SPMP were predicted using an ensemble voting approach (Supplementary Fig. 15B) with four different AMP prediction models: AMPscanner[74] (v2, convolutional neural network), AmpGram[75] (random forest model), AMPDiscover[76] (based on quantitative sequence activity models) and ABPDiscover (https://biocom-ampdiscover. cicese.mx/). Peptides predicted by antiSMASH in these RiPP BGCs were translated and all amino acid sequences with a length greater than 10 but lesser than 200 were used as inputs into these four models. Peptides were classified as AMPs if they received votes from both AMPscanner and AmpGram, and at least one vote from either AMP-Discover or ABPDiscover, and if the corresponding RiPP BGCs contained a transporter protein. The performance of this ensemble approach was evaluated using 78 known AMP sequences and 78 scrambled non-AMP sequences taken from the AmpGram benchmark dataset[75]. For our evaluation dataset, we identified and removed all sequences that were found in the training sets of AMPscanner, AmpGram, AMPDiscover and ABPDiscover using seqkit[77] (v0.11.0) and samtools faidx (v1.9). The percentage hydrophobicity and overall charge of selected peptide sequences was determined using the AMP calculator in the AMP database 3 (APD3; https://aps.unmc.edu/ prediction).

Out of 107 RiPP BGCs that were not bacteriocins, 54 of them were predicted to also be AMPs. One of these was found to be a lanthipeptide (GCF459) in *Dorea longicatena B* (Supplementary Fig. 15C), with no significant blastp matches to the NCBI nr database. This peptide sequence has a 32% hydrophobic amino acid composition and a net positive charge of +5, which could favor its insertion into the cell walls and membranes of its targets. Another novel AMP is a lasso-peptide (GCF26) found in a *Ruminococcus* species (Supplementary Fig. 15D), with similarly high proportion of hydrophobic amino acids (35%) and a slight net positive charge.

To associate BGC presence/absence patterns with microbial community structure, correlation analysis (Fastspar[78] v1.0.0, parameters: --iterations 100 --exclude_iterations 20, *p*-values from 1000 bootstrap replicates and permutation testing) was done based on SLC abundance profiles across samples (species with medium abundance ≤0.1% filtered out). Correlations in the network were kept if they had an associated *p*-value <0.05.

### Statistics and reproducibility

No statistical method was used to predetermine sample size. No data were excluded from the analyses. The experiments were not randomized. The Investigators were not blinded to allocation during experiments and outcome assessment.

### Reporting summary

Further information on research design is available in the Nature Research Reporting Summary linked to this article.

## Data availability

Source data are provided with this paper. Shotgun metagenomic sequencing data (Illumina and ONT) and SPMP hybrid MAGs are available from the European Nucleotide Archive (ENA – https://www. ebi.ac.uk/ena/browser/home) under project accession number PRJEB49168. All SPMP MAGs, a corresponding Kraken database, gene annotations and BGC sequences are available on Figshare at https:// figshare.com/collections/SPMP/5993596.

SPMP genomes were compared to the GTDB database (release 95, https://gtdb.ecogenomic.org). UHGG genomes and a corresponding Kraken database are available from http://ftp.ebi.ac.uk/pub/databases/ metagenomics/mgnify_genomes/human-gut/v1.0/. SGB genomes are available from http://segatalab.cibio.unitn.it/data/Pasolli_et_al.html. HRGM genomes from the initial release are available from https:// www.mbiomenet.org/HRGM/. The Kraken standard database used to assess *Bifidobacterium* abundances is available from https:// benlangmead.github.io/aws-indexes/k2. Databases used for anti-SMASH analysis of SPMP BGCs are available from v5.1.2 of the anti-SMASH software, while the MIBiG 2.0 database are available from https://mibig.secondarymetabolites.org/download. Source data are provided with this paper.

## Code availability

Source code for scripts used to analyze the data are available in a GitHub project at https://github.com/CSB5/SPMP[79].

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

## Acknowledgements
This work was supported by the A*STAR Computational Resource Center through the use of its high-performance computing facilities.

## Author contributions
Y.-Y.T., H.S., and N.N. conceived and designed the study. W.-Y.S. coordinated subject recall and collection of data/samples for the study under Y.-Y.T.'s supervision. A.L. and Y.D. coordinated DNA extraction with L.W.L.T.'s assistance and H.S.'s supervision. A.H.Q.N. coordinated library preparation and sequencing. D.B. performed initial analysis of data, including metagenomic assembly and comparison of short-read and hybrid assemblies under N.N.'s guidance. A.R. performed initial analysis of novel MAGs. J.-S.G. further refined the analysis and generated final results. M.C. performed BGC analysis and contributed to manuscript writing. J.-S.G. drafted the manuscript with inputs from A.H.Q.N. and N.N.'s edits. All authors provided further inputs and edits to the manuscript.

## Competing interests
The authors declare no competing interests.

## Additional information

[1]Genome Institute of Singapore, Singapore 138672, Singapore. [2]Life Sciences Institute, National University of Singapore, Singapore 117456, Singapore. [3]Baker Heart and Diabetes Institute, 75 Commercial Rd, Melbourne 3004 VIC, Australia. [4]Temasek Life Sciences Laboratory, 1 Research Link, Singapore 117604, Singapore. [5]Saw Swee Hock School of Public Health, National University of Singapore, 12 Science Drive 2, Singapore 117549, Singapore. [6]Department of Statistics and Applied Probability, National University of Singapore, Singapore 117546, Singapore. [7]Department of Biological Sciences, National University of Singapore, Singapore 117558, Singapore. [8]Yong Loo Lin School of Medicine, National University of Singapore, Singapore 117596, Singapore. [9]These authors contributed equally: Jean-Sebastien Gounot, Minghao Chia, Denis Bertrand. [10]These authors jointly supervised this work: Yik-Ying Teo, Henning Seedorf, Niranjan Nagarajan. ✉e-mail: yyteo@nus.edu.sg; henning@tll.org.sg; nagarajann@gis.a-star.edu.sg

