## [Peer Review File · Nature Communications]

Reviewer #1 (Remarks to the Author):

I thank the authors for addressing the points I raised.

Reviewer #2 (Remarks to the Author):

Thank you, the authors have addressed all of my comments and concerns!

Reviewer #3 (Remarks to the Author):

As the authors highlighted, this might be the first study to apply a hybrid approach (short and long-read sequencing technologies) to human gut metagenomes in a moderate-sized cohort (just over 100 healthy adults). As one would expect, this hybrid strategy could recover more MAGs (including novel ones) per sample, which are also more complete and accurate (less chimeric) than MAGs recovered using short reads only. There are clear advantages of using these hybrid MAGs as reference genomes, as the authors demonstrated, in improving metagenomic analyses (taxonomic classification and function annotation), when compared with reference collections populated with short-read MAGs (the status quo approach). The other main novelty is that the study population from Singapore is located in an under-represented geographical region (Southeast Asia), although it remains to be seen how representative and applicable the urban, Westernised Singaporean gut microbiome (SPMP catalog) is in relation to the much larger, diverse Southeast Asian populations.

In the revision, the authors have duly narrowed the focus to Southeast Asians, with apt comparison to the recent Asian population study (HRGM). Whilst I can understand the editorial rationale of using “South Asians” (previously Asian) in Title and Abstract, I still think “Singaporeans” would be more appropriate to describe the study (population), given the lack of sampling or analysis of any other major Southeast Asian populations (especially from non-urban lifestyles) in this study.

The authors also addressed my questions with respect to the MAG quality assessment. It is encouraging to see that the hybrid MAGs have a sizable reduction in MAG chimerism (as measured by GUNC, line 109-112). This is an important novel result highlighting the quality improvements of hybrid MAGs over short-read MAGs.

I also welcome the clearer highlighting of the like-for-like comparison of BGCs (just comparing antiSMASH hits, line 188-200). This will make the benefits of this work clearer for the reader to appreciate.

Finally, we applaud the authors for this important work on communicating the benefits of long-read sequencing technologies for constructing MAGs, the benefits of using population-specific reference databases, and for highlighting the exciting opportunities and importance of studying populations that are currently understudied.

Major issues:

1. I still have some issues with the claim that “Singaporean gut metagenomes harbour substantial uncharacterized diversity” (line 132 and the paragraph below).

I completely agree with the authors that, at the genome level, the study catalogue (SPMP) does harbour many novel species and strains (based on 99% ANI) compared to a public reference (UHGG), and the degree of discovery is substantially improved through hybrid sequencing. But at the metagenome level, the data presented (Supplementary Fig. 7A) don't really reflect “substantial uncharacterized diversity”, but rather very incremental (1%) species-level novelty not represented within the UHGG. This is an unimpressive result, considering the comparison is unfair to UHGG (de-replicated to species level whereas SPMP wasn't).

If the point (as the sub-heading indicates) is to demonstrate uncharacterized diversity in the Singaporean metagenomes at the species level (while strain-level comparison might not be computationally feasible), then the authors should perform a proper comparison at the species level, perhaps using the de-replicated sets of SPMP+UHGG versus UHGG alone, which would also demonstrate the value of population-specific reference genomes augmenting existing global catalogs.

In line with the above-mentioned comment on species-level analysis, the claim made here (Singaporean gut metagenomes are globally distinct at the genome/strain-level) cannot be directly supported by the comparison between the read classification rates with the full (non-dereplicated) UHGG and SPMP. A non-dereplicated genome database (i.e. SPMP, containing strain-level diversity) would be expected to classify more reads than a de-replicated database (UHGG, containing only species-level diversity). So this result (difference in the proportion of reads classified) may be better explained by this factor rather than due to the distinctness of Singaporean gut microbiomes.

Without performing additional strain-level mapping comparison, this statement (lines 143-145) and Supplementary Fig. 7B would be more appropriate to appear in Discussion rather than Results as a more speculative claim. Alternatively, a strain-level mapping against a subset of UHGG genomes (not the entire set) corresponding to SPMP species/SLCs should be computationally feasible and would be appropriate for showing uncharacterized strain-level diversity in Singaporean metagenomes.

2. the claim that “MAG quality can be better than isolate genome quality” in the paragraph referring to Fig. 1F-G, and “the quality of genomes that can be obtained using metagenomics is now comparable or better than what can be obtained from the sequencing of microbial isolates” in Discussion (line 247-250)

This is a bold claim not necessarily supported by the data included in this manuscript. Although MAGs *can* be higher quality than isolate genomes, it would be expected that isolate genomes would still be higher quality when the same sequencing technologies are used. Anything to the contrary has not been demonstrated in this work. This should be made clear here so that the reader is not inadvertently misled.

As it stands, the evidence presented for this is based on contiguity (N50, Fig. 1F) and completeness (MIMAG medium-to-high status change, Fig. 1G) against public (GTDB) genomes rather than the isolate genomes cultivated from the same sample to enable fair head-to-head comparisons (especially on chimerism). While I understand that the additional work required (WGS of cultivated isolates) would be outside the scope of the study, the authors should:

be clear that the isolate genomes being compared are public/GTDB and not from the same stool samples;

Be clear about what comparable/better “quality” actually refers to (i.e. contiguity/N50 and completeness/MIMAG, but not chimerism). This should be accompanied by a statement about the risk of chimerism in hybrid MAGs, indicating that there is more to genome quality than the contiguity and MIMAG when the benchmark is isolated genome (should be higher than among MAGs);

Acknowledge the limitations of MAGs compared to isolate genomes in Discussion.

Minor issues:

As per my previous comment, it would be helpful to have “hybrid” defined in Abstract and Introduction, just for a simple clarification on whether it refers to in this study (i.e. short/Illumina, long/Nanopore, and also Hi-C?). Although the word “hybrid” shouldn’t be foreign to the microbial (meta)genomics field, re-introducing this concept should make the paper more accessible to the readership of Nature Communications.

Line 126 – 128 - needs actual numbers to back up the statement on how “commonly used” long-read is used for generating isolate genomes in GTDB. It might be better to remove this sentence unless the authors could stratify the Fig1-G comparisons against isolate genomes by short and long-read, respectively.

Line 132 - This title should be focused on “Singaporean” or “Southeast Asian”, not “Asian”, since this is what the contents of this section speak to.

Line 139-143 - Along with comparing the size of the UHGG and SPMP reference genome databases, the degree of dereplication should also be compared as they are not really comparable. The authors should make it clear that the UHGG database used has been de-replicated to the species level, whereas SPMP is not, and so contains strain-level diversity.

Line 143-145 - This sentence needs clarification. Is this sentence saying that the independent Singaporean cohorts are similar to each other but distinct within global diversity? That is the only meaning I could come up with, but the language is unclear.

Figures:

Fig. 1C

I still find the axis labels confusing. If I understand correctly, the x- and y-axis show the Kraken2/Bracken relative abundance (of 109 samples/points) derived from SPMP custom databases (one based on hybrid MAGs, the other based on short-read MAGs) versus the Kraken2/Bracken relative abundance derived from the default (clarify if it is UHGG?) Kraken-DB genomes database? If so, “default Kraken-DB” need to be reflected in the x-axis label. If the y-axis relative abundance is not

Kraken2/Bracken estimation, could the authors please provide more details of this analysis in Methods?

This figure is referred to by three claims made between lines 101-104, which all require actual numbers and statistics to back up. For instance, 1) how 'substantial' was the improvement in read assignment on Bifidobacterium genomes? 2) how much fewer genomes were detected when using short-reads MAGs? 3) how consistent (when compared with default/UHGG database?) was abundance estimation based on hybrid MAGs versus short-read MAGs?

The other confusing part is the significant number of points/samples scattered along with the x-axis at the bottom of Fig.1C, could these be false-positive species/genomes (undetected by hybrid sequencing) likely misclassified by the default Kraken database? To simplify the core message, could the authors provide clarification on these outliers in the figure caption or/and filter them out with a relative abundance cut-off?

Fig 1F-G would benefit from further clarifications and providing Source Data, specifically:

Fig.1F Y-axis label should clearly state improvement in contiguity (N50). It would also be beneficial to label on top the % of representative species whose reference genomes are improved/replaced with SPMP over GTDB.

Fig. 1G highlights around 40-50 'incomplete' (medium MIMAG quality) isolate genomes in GTDB, some of which may reflect the mislabelling issues known in GTDB (i.e. MAGs submitted to GenBank as isolates). Could the authors please double-check the primary source of these GTDB isolate genomes (i.e. if they are genuinely isolates, as previously commented by Referee #1), then provide the Source Data (GTDB accessions) as a Supplementary Table. This list would benefit the microbial genomics community and GTDB curators so that certain low-quality or mislabelled isolate genomes in GTDB could be investigated further or replaced/avoided.

REVIEWER COMMENTS

Rev #1: microbiome, bioinformatics.

Rev #2: microbiome, bioinformatics, sequencing technologies.

Rev #3: microbial genomics, microbiome.

We are pleased to note that our last revision addressed all comments from reviewer 1 and 2. We have now carefully addressed the remaining comments from reviewer 3 in this revised version, as detailed below.

Reviewer #1 (Remarks to the Author):

I thank the authors for addressing the points I raised.

Reviewer #2 (Remarks to the Author):

Thank you, the authors have addressed all of my comments and concerns!

Reviewer #3 (Remarks to the Author):

As the authors highlighted, this might be the first study to apply a hybrid approach (short and long-read sequencing technologies) to human gut metagenomes in a moderate-sized cohort (just over 100 healthy adults). As one would expect, this hybrid strategy could recover more MAGs (including novel ones) per sample, which are also more complete and accurate (less chimeric) than MAGs recovered using short reads only. There are clear advantages of using these hybrid MAGs as reference genomes, as the authors demonstrated, in improving metagenomic analyses (taxonomic classification and function annotation), when compared with reference collections populated with short-read MAGs (the status quo approach). The other main novelty is that the study population from Singapore is located in an under-represented geographical region (Southeast Asia), although it remains to be seen how representative and applicable the urban, Westernised Singaporean gut microbiome (SPMP catalog) is in relation to the much larger, diverse Southeast Asian populations.

In the revision, the authors have duly narrowed the focus to Southeast Asians, with apt comparison to the recent Asian population study (HRGM). Whilst I can understand the editorial rationale of using "South Asians" (previously Asian) in Title and Abstract, I still think "Singaporeans" would be more appropriate to describe the study (population), given the lack of sampling or analysis of any other major Southeast Asian populations (especially from non-urban lifestyles) in this study.

Response: We would like to clarify that we are not suggesting that our data represents all Southeast Asians, just as databases like HRGM are not likely to properly represent all of Asia given the tremendous diversity that is present here. As the reviewer noted, our study provides data for an understudied region (Southeast Asia) which is what we are trying to highlight. Our cohort includes subjects from three ethnicities, two of which represent significant populations in Southeast Asia (Malay and Chinese) and this is another strength of our work. We have now clarified these points in the manuscript.

The authors also addressed my questions with respect to the MAG quality assessment. It is encouraging to see that the hybrid MAGs have a sizable reduction in MAG chimerism (as measured by GUNC, line 109-112). This is an important novel result highlighting the quality improvements of hybrid MAGs over short-read MAGs.

I also welcome the clearer highlighting of the like-for-like comparison of BGCs (just comparing

antiSMASH hits, line 188-200). This will make the benefits of this work clearer for the reader to appreciate.

Finally, we applaud the authors for this important work on communicating the benefits of long-read sequencing technologies for constructing MAGs, the benefits of using population-specific reference databases, and for highlighting the exciting opportunities and importance of studying populations that are currently understudied.

Response: We thank the reviewer for these encouraging comments and for recognizing the value of our work.

Major issues:

1. I still have some issues with the claim that “Singaporean gut metagenomes harbour substantial uncharacterized diversity” (line 132 and the paragraph below).

I completely agree with the authors that, at the genome level, the study catalogue (SPMP) does harbour many novel species and strains (based on 99% ANI) compared to a public reference (UHGG), and the degree of discovery is substantially improved through hybrid sequencing. But at the metagenome level, the data presented (Supplementary Fig. 7A) don’t really reflect “substantial uncharacterized diversity”, but rather very incremental (1%) species-level novelty not represented within the UHGG. This is an unimpressive result, considering the comparison is unfair to UHGG (de-replicated to species level whereas SPMP wasn’t).

Response: We would like to clarify that all claims of microbial genetic diversity were meant at the “genome level” and as the reviewer agrees, we have demonstrated that. Our read mapping/classification analysis with SPMP was only to show utility for this function and not at the “metagenome level” as the reviewer assumed. We have now made this point clear in the manuscript through a summary of the results in the first paragraph (“... better references for microbiome read mapping or taxonomic classification ...”). For mapping/classification analysis it is important to consider computational feasibility and what can be achieved with similar resources. Therefore our comparisons with similar sized databases should be appropriate.

If the point (as the sub-heading indicates) is to demonstrate uncharacterized diversity in the Singaporean metagenomes at the species level (while strain-level comparison might not be computationally feasible), then the authors should perform a proper comparison at the species level, perhaps using the de-replicated sets of SPMP+UHGG versus UHGG alone, which would also demonstrate the value of population-specific reference genomes augmenting existing global catalogs.

Response: We believe the reviewer has made unintended inferences from the sub-heading and we have tried to avoid that. The subheading now reads “... microbial genetic diversity”, to be clear that the attribute “substantial” refers to microbial genetics and not to the metagenome/microbiome. We have also replaced metagenome with microbiome as that may be less confusing.

In line with the above-mentioned comment on species-level analysis, the claim made here (Singaporean gut metagenomes are globally distinct at the genome/strain-level) cannot be directly supported by the comparison between the read classification rates with the full (non-dereplicated) UHGG and SPMP. A non-dereplicated genome database (i.e. SPMP, containing strain-level diversity) would be expected to classify more reads than a de-replicated database (UHGG, containing only species-level diversity). So this result (difference in the proportion of reads classified) may be better explained by this factor rather than due to the distinctness of Singaporean gut microbiomes.

Without performing additional strain-level mapping comparison, this statement (lines 143-145) and Supplementary Fig. 7B would be more appropriate to appear in Discussion rather than Results as a more speculative claim. Alternatively, a strain-level mapping against a subset of UHGG genomes (not the entire set) corresponding to SPMP species/SLCs should be computationally feasible and would be appropriate for showing uncharacterized strain-level diversity in Singaporean metagenomes.

Response: Please refer to our point above that we are not using this analysis to argue for “uncharacterized microbial genetic diversity”. Instead we are just trying to show utility of high-quality population-specific references. The rest of the paragraphs in this section are indeed meant to support the claims on “microbial genetic diversity”.

Nevertheless, we have tried to do an experiment in the spirit of what the reviewer has suggested. Specifically, we compared strain-level mapping using SPMP vs UHGG genomes with species-specific databases containing the same number of strains (**Supplementary Note 2**). We observed increased sensitivity for mapping short reads from Singaporean gut metagenomes to SPMP vs UHGG strain genomes, particularly for abundant gut bacteria such as *Prevotella copri* and *Alistipes onderdonkii* (**Supplementary Note 2**).

2. the claim that “MAG quality can be better than isolate genome quality” in the paragraph referring to Fig. 1F-G, and “the quality of genomes that can be obtained using metagenomics is now comparable or better than what can be obtained from the sequencing of microbial isolates” in Discussion (line 247-250)

This is a bold claim not necessarily supported by the data included in this manuscript. Although MAGs *can* be higher quality than isolate genomes, it would be expected that isolate genomes would still be higher quality when the same sequencing technologies are used. Anything to the contrary has not been demonstrated in this work. This should be made clear here so that the reader is not inadvertently misled.

As it stands, the evidence presented for this is based on contiguity (N50, Fig. 1F) and completeness (MIMAG medium-to-high status change, Fig. 1G) against public (GTDB) genomes rather than the isolate genomes cultivated from the same sample to enable fair head-to-head comparisons (especially on chimerism). While I understand that the additional work required (WGS of cultivated isolates) would be outside the scope of the study, the authors should: be clear that the isolate genomes being compared are public/GTDB and not from the same stool samples; Be clear about what comparable/better “quality” actually refers to (i.e. contiguity/N50 and completeness/MIMAG, but not chimerism). This should be accompanied by a statement about the risk of chimerism in hybrid MAGs, indicating that there is more to genome quality than the contiguity and MIMAG when the benchmark is isolated genome (should be higher than among MAGs); Acknowledge the limitations of MAGs compared to isolate genomes in Discussion.

Response: We agree with the reviewer that a direct comparison between isolates genomes and MAGs from the same sample would be ideal. We also agree that there are other measures of assembly quality. We have therefore made the metrics that we have used explicit in the text, and made the comparison more concrete (i.e. GTDB isolates and not isolates in general). We have also highlighted that a substantial proportion of GTDB genomes are based on short reads.

Minor issues:

As per my previous comment, it would be helpful to have “hybrid” defined in Abstract and Introduction, just for a simple clarification on whether it refers to in this study (i.e. short/Illumina, long/Nanopore, and also Hi-C?). Although the word “hybrid” shouldn’t be foreign to the microbial

(meta)genomics field, re-introducing this concept should make the paper more accessible to the readership of Nature Communications.

Response: As suggested, we have clarified what we mean by “hybrid” in the abstract and introduction.

Line 126 – 128 - needs actual numbers to back up the statement on how “commonly used” long-read is used for generating isolate genomes in GTDB. It might be better to remove this sentence unless the authors could stratify the Fig1-G comparisons against isolate genomes by short and long-read, respectively.

Response: We have reworded the sentence to avoid the reference to “commonly used” and long-read genomes.

“While the improvement in assembly is expected for uncultivated organisms that are primarily assembled using short-read metagenomics, the observed improvement relative to GTDB isolates (albeit smaller, Wilcoxon p-value= 1.25×10^{-11}) is noteworthy as assembling the latter is typically less error prone.”

Line 132 - This title should be focused on “Singaporean” or “Southeast Asian”, not “Asian”, since this is what the contents of this section speak to.

Response: We had meant this title from the perspective that “Southeast Asians are Asians” and finding substantial novel microbial genetic diversity in this population highlights the need to characterize other understudied Asian populations. We therefore think this title is appropriate and would like to keep it as such.

Line 139-143 - Along with comparing the size of the UHGG and SPMP reference genome databases, the degree of dereplication should also be compared as they are not really comparable. The authors should make it clear that the UHGG database used has been de-replicated to the species level, whereas SPMP is not, and so contains strain-level diversity.

Response: We have made this distinction more explicit now in the manuscript, stating that the SPMP database is at strain-level while the UHGG database is at species-level.

Line 143-145 - This sentence needs clarification. Is this sentence saying that the independent Singaporean cohorts are similar to each other but distinct within global diversity? That is the only meaning I could come up with, but the language is unclear.

Response: We would like to clarify again that the point here is not about diversity and instead about utility for read classification. For this purpose, we did want to show that population-specific databases improve over “generic” databases. We have reworded this text now to make it clearer.

Figures:

Fig. 1C

I still find the axis labels confusing. If I understand correctly, the x- and y-axis show the Kraken2/Bracken relative abundance (of 109 samples/points) derived from SPMP custom databases (one based on hybrid MAGs, the other based on short-read MAGs) versus the Kraken2/Bracken relative abundance derived from the default (clarify if it is UHGG?) Kraken-DB genomes database? If so, “default Kraken-DB” need to be reflected in the x-axis label. If the y-axis relative abundance is not Kraken2/Bracken estimation, could the authors please provide more details of this analysis in Methods?

Response: This is indeed what is plotted in this figure. We have renamed the x-axis label and edited the figure description to define the database used in this analysis (standard kraken database, available on the software's website).

This figure is referred to by three claims made between lines 101-104, which all require actual numbers and statistics to back up. For instance, 1) how 'substantial' was the improvement in read assignment on *Bifidobacterium* genomes? 2) how much fewer genomes were detected when using short-reads MAGs? 3) how consistent (when compared with default/UHGG database?) was abundance estimation based on hybrid MAGs versus short-read MAGs?

Response: We would like to clarify that there are really only two claims, where the first one ("substantial improvement in read assignment") was meant as a summary of the other two. We now include a new supplementary figure (#3) to expand on results represented in **Figure 1C**, i.e. that more genomes were detected using hybrid genomes (n=217 vs n=119), and that the median relative error in relative abundance was less using hybrid genomes (8% vs 73%, relative to standard Kraken database).

The other confusing part is the significant number of points/samples scattered along with the x-axis at the bottom of Fig.1C, could these be false-positive species/genomes (undetected by hybrid sequencing) likely misclassified by the default Kraken database? To simplify the core message, could the authors provide clarification on these outliers in the figure caption or/and filter them out with a relative abundance cut-off?

The points found along the x-axis represent *Bifidobacterium* species found using the Kraken standard database but not found within either and mostly in short-reads MAGs or (less likely) in hybrid MAGs. We modified the legend for **Figure 1** to clearly describe this part.

Fig 1F-G would benefit from further clarifications and providing Source Data, specifically:

Fig.1F Y-axis label should clearly state improvement in contiguity (N50). It would also be beneficial to label on top the % of representative species whose reference genomes are improved/replaced with SPMP over GTDB.

Response: We modified the figure label and added the requested percentages in the figure.

Fig. 1G highlights around 40-50 'incomplete' (medium MIMAG quality) isolate genomes in GTDB, some of which may reflect the mislabelling issues known in GTDB (i.e. MAGs submitted to GenBank as isolates). Could the authors please double-check the primary source of these GTDB isolates genomes (i.e. if they are genuinely isolates, as previously commented by Referee #1), then provide the Source Data (GTDB accessions) as a Supplementary Table. This list would benefit the microbial genomics community and GTDB curators so that certain low-quality or mislabelled isolate genomes in GTDB could be investigated further or replaced/avoided.

Response: We now provide an additional table (**Supplementary file 4**) containing GTDB accession numbers for both isolate and uncultivated SLCs which are improved from medium to high with SPMP genomes. For all accessions, we extracted NCBI biosample information to validate isolate origin and added this information to the supplementary file.

REVIEWERS' COMMENTS

Reviewer #3 (Remarks to the Author):

I commend and thank the authors' efforts for addressing all of my comments.

A couple of final suggestions on data release:

While I appreciate that the raw data and scripts of this study have already been made public, it would be even better if the authors would also make the SPMP genomes, annotations and database files (i.e. Kraken, BGCs) available on a public repository (similar to that of the UHGG and HRGM).

I also strongly encourage the authors to submit the SPMP SLC representative MAGs to GenBank/ENA so the improved and novel genomes from this important work would be automatically incorporated in future releases of the global microbial genome databases (e.g. UHGG, GTDB). Making these resources available to the public would be extremely useful to the microbiome research community, particularly to those studying Southeast Asian populations with limited computational resources.

Reviewer #3 (Remarks to the Author):

I commend and thank the authors' efforts for addressing all of my comments.

A couple of final suggestions on data release:

While I appreciate that the raw data and scripts of this study have already been made public, it would be even better if the authors would also make the SPMP genomes, annotations and database files (i.e. Kraken, BGCs) available on a public repository (similar to that of the UHGG and HRGM).

We provided SPMP genomes, annotations and database files on FigShare (<https://figshare.com/collections/SPMP/5993596>).

I also strongly encourage the authors to submit the SPMP SLC representative MAGs to GenBank/ENA so the improved and novel genomes from this important work would be automatically incorporated in future releases of the global microbial genome databases (e.g. UHGG, GTDB). Making these resources available to the public would be extremely useful to the microbiome research community, particularly to those studying Southeast Asian populations with limited computational resources.

We submitted the hybrid assemblies into ENA under the same project accession number PRJEB49168.